# Ferroptosis in Toxicology: Present and Future

**DOI:** 10.3390/ijms26146658

**Published:** 2025-07-11

**Authors:** Birandra K. Sinha

**Affiliations:** Mechanistic Toxicology Branch, Division of Translational Toxicology, National Institute of Environmental Health Sciences, NIH, Research Triangle Park, Durham, NC 27709, USA; sinha1@niehs.nih.gov

**Keywords:** ferroptosis, environmental pollutants, PFAS, heavy metals, pesticides

## Abstract

Ferroptosis, a regulated form of cell death characterized by iron-dependent lipid peroxidation, has emerged as a pivotal mechanism in understanding the toxicological effects of various environmental pollutants. This short review delves into the intricate pathways of ferroptosis, its induction by diverse environmental toxicants, and the subsequent implications for human health. By elucidating and understanding pathways involved in environmental exposures and ferroptosis, we aim to shed light on potential therapeutic interventions and preventive strategies. Furthermore, identifications of biomarkers of ferroptosis will aid in monitoring ferroptosis-induced diseases/tissue damage, promoting the development of targeted therapies.

## 1. Introduction

Ferroptosis is characterized by an iron-dependent accumulation of lethal reactive oxygen species (ROS) and lipid radicals. The most frequently occurring free radicals and reactive molecules in biological systems are derived from oxygen (ROS) and nitrogen (Reactive Nitrogen Species, RNS). ROS or RNS are formed during electron transfer reactions, by losing or accepting electron(s) [1,2]. While low levels of ROS are important for cellular proliferation, differentiation, migration, apoptosis, and necrosis, excessive ROS causes oxidative stress, inducing ROS-mediated damage to cellular biomolecules including DNA, proteins, and membranes [3,4]. Oxidative stress is characterized by a shift in equilibrium between the formation and elimination of free radicals. Oxidative stress is involved in various aspects of human pathogenesis in chronic diseases, aging and cancers [5]. Human normal as well as tumor cells have an enhanced antioxidant system to remove/detoxify ROS formed. They include superoxide dismutase (SOD), catalase (CAT) and glutathione peroxidases (GPXs), as well as reduced glutathione [2] to protect cells from ROS-induced toxicity [6]. Excess ROS and lipid ROS formation cause damage to cellular membranes inducing ferroptosis, a form of cell death which is morphologically, biochemically, and genetically distinct from apoptosis, necrosis, and autophagy [7,8,9]. As ferroptosis is distinct from other forms of cell death it has been suggested to play an important role in cancer therapy.

Glutathione peroxidase 4 (GSH/GPX4) and system Xc^−^, are two major signaling pathways which are known to play pivotal roles in ferroptosis [7]. System Xc^−^ is the family of heterodimeric amino acid transporters and SLC7A11 functions to exchange L-cystine and L-glutamate [10]. Erastin, a ferroptosis inducer, inhibits the function of system Xc^−^ and depletes glutathione (GSH), resulting in inhibition of GPX4 and the accumulation of ROS and lipid peroxides which in turn causes ferroptosis.

Ferroptosis has recently been suggested to play a significant role in toxicology due to its unique cell death mechanism that is driven by iron-dependent lipid peroxidation. This mechanism has become extremely relevant to toxicology because exposure to certain toxins (e.g., heavy metals, environmental pollutants, and industrial chemicals) can intensify oxidative stress, leading to ferroptotic cell damage (and death) in organs like the liver, kidneys, and lungs [11]. Many drugs, especially chemotherapy agents, have been shown to increase ROS formation and disrupt cellular iron metabolism, causing ferroptosis [12]. This is shown in Figure 1.

Ferroptosis has been implicated in neurotoxicity due to its role in oxidative damage and iron accumulation, which are common in neurodegenerative diseases like Alzheimer’s and Parkinson’s [13]. Since some neurotoxic substances can elevate iron or lipid peroxidation in the brain, insights into ferroptosis could lead to new protective strategies for managing neurotoxin exposure and associated diseases.

The liver, as a central organ in detoxification, is highly susceptible to ferroptosis [14]. This is especially relevant for drugs that induce iron overload and/or oxidative stress, triggering ferroptosis and causing hepatotoxicity and liver fibrosis.

Exposure to environmental toxins, such as heavy metals (e.g., arsenic, cadmium) and other industrial pollutants, can influence cellular iron metabolism or redox balance, increasing the risk of ferroptosis in exposed tissues [15,16]. Understanding the mechanisms and pathways involved in ferroptosis will help in predicting, preventing, or treating drug-induced tissue damage, especially in high-risk organs. Research on ferroptosis can then provide strategies to mitigate toxic effects and improve public health outcomes. By targeting ferroptosis pathways, researchers can significantly reduce toxicity and prevent cell death from toxic xenobiotics or drugs. Ferroptosis inhibitors may then be utilized to mitigate damage in ferroptosis-sensitive tissues/organs during exposures or poisonings from environmental pollutants, providing a new avenue for therapeutic intervention in toxicology.

### 1.1. Perfluoroalkyl and Polyfluoroalkyl Substances (PFAS)

PFAS are a group of persistent chemicals that are widely present in environment due to their extensive use as protective coatings for carpets and clothing, water repellents, paper coatings, and surfactants [17,18,19]. PFAS are now found in water and food, as well as in indoor air settings, leading to significant exposures [18]. Unfortunately, efforts to control and decrease the use of PFAS have not been very successful as they are still detected in humans, animals, and water due to both their extensive world-wide application and lack of biodegradability. Air pollution and cigarette smoke have been shown to contain high concentrations of PFOS, a key member of PFAS family. PFAS causes oxidative stress [20,21], mitochondrial dysfunction [22], and inhibition of gap junction intercellular communication [23], resulting in serious human health problems including endocrine disruptions, liver damage, kidney cancer, obesity, hypertension, and immunotoxicity [24].

In vitro and in vivo experiments, as well as population studies, have suggested that exposure to PFAS is associated with increased oxidative stress. PFAS generate reactive ROS in human hepatocytes, inducing oxidative stress and causing DNA damage with genotoxic and cytotoxic potential [25]. In addition, PFAS exposures have been shown to enhance oxidative stress in kidney, leading to disruptive effects on the peroxisome proliferator-activated receptor (PPAR) and its downstream functions [21]. Several studies in humans have shown that serum PFAS concentrations are strongly associated with biomarkers of oxidative and nitrative stress as PAFS increase the levels of both 8-nitroguanine and 8-hydroxy-2-deoxyguanosine (8-OHdG) [26]. Furthermore, in a randomized control trial in senior Koreans with elevated serum levels of perfluorododecanoic acid (PFDoDA), and PFOS blood concentrations showed higher levels of malondialdehyde and 8-OHdG in urine [27].

PFAS is known to accumulate in the kidneys and research indicates a strong link between exposure to PFAS with kidney toxicity, showing that PFAS can negatively impact kidney function and potentially contribute to the development of chronic kidney disease. Further studies have reported that PFAS exposure induces oxidative stress in the kidney, causing lipid peroxidation and DNA damage, increased production of mitochondrial transport chain proteins, decreased cell proliferation, and apoptosis [21,28,29,30]. Multiple pathways have been implicated in PFOS exposure to cause renal damage, including dysregulation of PPARa and PPARg, key nuclear receptor hormones. These are highly expressed in the proximal renal tubule and involved in lipid metabolism, lipogenesis, glucose homeostasis, and cell growth and differentiation [21]. Recently, the use of N-acetylcysteine (NAC) and taurine—both known inhibitors of oxidative stress—has been found to reduce complications in diabetic and hypertensive patients with chronic kidney disease [31].

PFAS can cross the blood–brain barrier and accumulate in the brain which negatively impacts brain function and disrupt neurotransmitter systems, potentially leading to cognitive impairments and neurodevelopmental issues [32,33]. This is particularly important when exposure occurs during critical developmental stages since PFAS can cross-disrupt various mechanisms within the brain, including calcium homeostasis and mitochondrial function and alterations in neurotransmitters of neurons [34].

PFAS exposure has been shown to disrupt the cellular antioxidant balance by decreasing the activity of GPX4, a key enzyme responsible for preventing lipid peroxidation and ferroptosis. While the precise mechanism of an inhibitory effect of PFAS on GPX4 is currently not known, it appears to be multifactorial, involving indirect suppression via GSH depletion, oxidative and epigenetic modifications of GPX4 as well as direct binding interactions. Understanding these mechanisms is critical, as GPX4 inhibition is central to ferroptotic cell death, linking PFAS exposure to inflammation, organ damage, and chronic disease development. Further research using molecular modeling, proteomics, and functional assays is needed to fully define these interactions.

Studies have found elevated intracellular iron levels, increased lipid peroxidation, and altered expression of ferroptosis-related proteins like ACSL4 (acyl-CoA synthetase long-chain family member 4) when cells are exposed to PFAS.

### 1.2. Heavy Metals

Heavy metals (e.g., arsenic, cadmium, and mercury) are found in the Earth’s crust with varying concentrations and are not easily detoxified into less toxic compounds through metabolic processes. Thus, they persist for a long time and are taken up by plants, leading to accumulation in various animals and humans at high toxic levels, causing harmful impacts on both the environment and human health [35,36]. 

Many heavy metals can interfere with the synthesis or function of glutathione, a crucial antioxidant that protects against lipid peroxidation, thereby promoting ferroptosis. Copper has been reported to inhibit GPX4 activity, leading to its degradation through autophagy and promoting ferroptosis [37]. GPX4 is the key enzyme for the removal of cellular lipid peroxide formed, inhibiting ferroptosis. Heavy metals disrupt mitochondrial function, leading to increased ROS production and further promoting ferroptosis [15]. Exposure to heavy metals, especially lead and mercury, are linked to neurodegenerative diseases like Alzheimer’s and Parkinson’s. Methylmercury (CH_3_-Hg) has been shown to react with GSH, generate ROS and cause oxidative stress Ferroptosis has been suggested as a potential mechanism of neuronal death [13,16]. The combination of metals like iron with carbon tetrachloride can cause liver damage through ferroptosis [38]. 

Heavy metals have been reported to disrupt cellular redox homeostasis and contribute to iron dysregulation, promoting ferroptosis [39]. Cadmium exposure can induce iron accumulation in cells, enhancing the production of ROS and lipid peroxidation, inducing ferroptosis [40]. Thus, ferroptosis in response to heavy metal exposure primarily impacts organs with high iron content or metabolic activity, such as the liver and kidneys [14,38]. Exposure to arsenic can cause ferroptosis in liver cells, contributing to arsenic-induced liver injury and increasing the risk of hepatotoxicity and fibrosis [41].

### 1.3. Industrial Pollutants and Airborne Particulates

Particulate matter (PM) is the principal component of indoor and outdoor air pollution. PM includes a range of particle sizes, such as coarse, fine, and ultrafine particles. Industrial pollutants, especially airborne particulate matter from sources like vehicle exhaust and industrial emissions, contain metals and organic compounds [42,43,44]. Particles that are <100 nm in diameter are defined as ultrafine particles (UFPs). Chronic exposure to air pollutants that induce ferroptosis can damage lung tissue, leading to chronic obstructive pulmonary disease (COPD) and lung cancer [45]. When inhaled, these particles can produce oxidative stress in lung cells, causing ferroptosis and leading to respiratory diseases. Inhalation of PM exacerbates respiratory symptoms in patients with chronic airway disease, but the mechanism for this response is not clear currently.

### 1.4. Pesticides

Highly lipophilic pesticides are known to integrate into cell membranes and initiate lipid peroxidation which in the presence of iron lead to ferroptosis [46]. Paraquat, a pesticide and a ROS generator [47,48], has been shown to induce ferroptosis in neural and other cell types, leading to neurodegenerative conditions like Parkinson’s disease [49,50]. Ferroptosis induced by pesticides exposure poses significant neurotoxic risks, as brain cells are highly susceptible to oxidative damage and lipid peroxidation, elucidating why long-term exposure to certain pesticides has been associated with increased risks of neurological disorders [51,52].

Imidacloprid (IMI) is among the common neonicotinoid insecticides used in agriculture worldwide, posing a potential toxic threat to non-target animals and humans. Numerous studies have shown that ferroptosis is involved in the pathophysiological progression of renal diseases [53,54]. Chlorpyrifos is an organophosphate pesticide that can trigger ferroptosis, causing liver toxicity [55]. Interestingly, Vitamin E (a strong antioxidant) supplementation has been reported to help reverse the effects of ferroptosis caused by chlorpyrifos [56]. Various other pesticides, e.g., Acetochlor, an herbicide that causes skin irritation, cancer, and developmental issues, has been reported to trigger ferroptosis [57].

### 1.5. Ferroptosis Combined with Apoptosis and Necrosis

Exposure to environmental pollutants can also cause other forms of cell death such as apoptosis and necrosis [29,58,59]. Ferroptosis can act synergistically with apoptosis, significantly enhancing tissue damage. Thus, simultaneous exposure to multiple pollutants, may increase ferroptotic damage, especially when both iron-dependent mechanisms and lipid peroxidation are enhanced. Cells exposed to chronic low doses of environmental pollutants may initially activate protective mechanisms (like antioxidant defenses) to counteract ferroptosis. However, prolonged exposure can overwhelm these defenses, tipping the balance toward ferroptosis, which could explain why chronic, low-level pollutant exposure is linked to long-term health risks.

### 1.6. Ferroptosis and Apoptosis Crosslinks

Ferroptosis and apoptosis are historically considered distinct, but studies indicate inter-pathway communication, especially during pollutant-driven oxidative stress. Lipid peroxidation and mitochondrial dysfunction resulting from environmental pollutants (PFAS, Heavy metals, etc.) exposure, mitochondrial membrane may become destabilized, leading to cytochrome c release, a hallmark of apoptosis. The tumor suppressor protein p53 plays a pivotal role in both apoptosis and ferroptosis. In apoptosis, p53 promotes the expression of pro-apoptotic genes such as *BAX* and *PUMA*, leading to mitochondrial outer membrane permeabilization and caspase activation. In contrast, p53 can repress SLC7A11 [60], a component of the system Xc^−^, leading to a reduction in glutathione synthesis, decreasing GPX4 activity, enhancing lipid peroxidation and inducing ferroptosis. Members of the BCL-2 family, particularly BID and BAX, are central to the intrinsic apoptotic pathway. Truncated BID (tBID), generated by caspase-8 cleavage, is translocated to mitochondria, promoting cytochrome c release and apoptosis. Interestingly, tBID also has been shown to contribute to mitochondrial dysfunction and ROS production and influencing ferroptotic sensitivity [61]. In this regard, DNA damage caused by pollutants has been reported to induce Bax and Bak activation, cytochrome c release, caspases activation, and necrotic cell death in cells [62].

Co-exposure of Benzo[a]pyrene (BaP) and Arsenic has been reported to induce lipid peroxidation and Nrf2 suppression, reducing antioxidant defense [63]. Additionally, they triggered DNA damage response, p53 activation, and increased cleaved caspase-3 expression [64]. Ferroptotic inhibition with ferrostatin-1 reduced both ferroptotic and apoptotic markers, highlighting pathway overlap.

While both apoptosis and ferroptosis involve reactive oxygen species (ROS), their sources and targets differ. Apoptosis is driven primarily by mitochondrial ROS and cytochrome c release, while ferroptosis is characterized by iron-dependent lipid peroxidation, facilitated by lipoxygenases and ACSL4-mediated incorporation of polyunsaturated fatty acids into phospholipids. Endoplasmic reticulum (ER) stress is another shared upstream event between apoptosis and ferroptosis. The unfolded protein response can promote apoptosis through CHOP induction and JNK activation and contributes to ferroptosis by disturbing lipid metabolism and redox homeostasis [65].

The decision between apoptosis and ferroptosis is influenced by several factors such as cellular redox balance, antioxidant capacity, iron availability and caspase activation. It has been shown that high intracellular glutathione levels and GPX4 activity favor survival or apoptosis, while their depletion results in ferroptosis. Similarly, elevated labile iron pools cause susceptibility to ferroptosis by enhancing lipid peroxidation, with limited or no impact on apoptotic pathway. Furthermore, caspase activation leads to apoptosis without affecting the ferroptotic pathway.

### 1.7. Ferroptosis–Necrosis Crosslinks

Ferroptosis and necrosis also share functional crosslinks, primarily through oxidative stress, lipid membrane damage, and inflammation. Ferroptosis and necroptosis are alternative, in that resistance to one pathway sensitized cells to death via the other pathway [66]. Necrosis has been reported to involves RIPK1–RIPK3–MLKL pathway, resulting in membrane rupture and the release of damage-associated molecular patterns (DAMPs) [67]. While distinct from ferroptosis, oxidative stress and iron metabolism modulate both processes. For instance, RIPK1 has been shown to influence intracellular iron trafficking, potentially linking necroptosis to ferroptotic sensitivity.

Pollutants like arsenic or PM2.5 produce mitochondrial ROS and activate RIPK1/RIPK3/MLKL necroptotic pathways, elevating lipid peroxidation, enhancing ferroptotic vulnerability [68,69]. Iron accumulation from heavy metals can cause lysosomal rupture and necrotic cell death, synergizing with iron-catalyzed lipid peroxidation in ferroptosis.

Factors influencing the decision between ferroptosis, and necrosis include many factors, including energy status. Ferroptosis is a regulated, ATP-independent process, whereas necrosis often occurs under conditions of profound ATP depletion. Induction of ferroptosis is tightly dependent upon intracellular iron levels and lipid ROS, while necrosis may result from broader oxidative or physical damage. In addition, necrosis is suggested to be triggered by calcium overload, leading to calpain activation and membrane breakdown; similar calcium-related stress can sensitize cells to ferroptosis via mitochondrial dysfunction.

The decision between ferroptosis, apoptosis, and necrosis arises from a delicate balance between damage signals, and regulatory checkpoints. Crosslinks among these pathways allow for compensatory or alternative death execution when one pathway is blocked—for instance, caspase inhibition can divert cells from apoptosis toward ferroptosis or necroptosis. It appears that multi-pollutant co-exposure activates a complex interplay of ferroptosis, apoptosis, and necrosis through shared molecular mediators like ROS, p53, mitochondrial damage, lipid peroxidation, and iron dysregulation. This integrated network leads to synergistic cytotoxic effects, underscoring the importance of a systems-level approach in environmental toxicology. More studies are needed to precisely map these death pathway intersections and their role in pollutant-induced disease. Understanding these regulatory networks will clarify the molecular pathways leading to cell death and will also provide new therapeutic avenues in diseases characterized by dysregulated cell death, such as cancer, neurodegeneration, and ischemia–reperfusion injury.

### 1.8. Biomarkers of Ferroptosis

Biomarkers are important tools for the early detection of tissue damage in any biological process, in the treatment of cancers by chemotherapy or exposure to environmental toxins. Thus, the identification of biomarkers of ferroptosis can significantly enhance the field of toxicology by providing tools to monitor, predict, and potentially mitigate cell death related to iron-dependent lipid peroxidation. Furthermore, biomarkers of ferroptosis can be advantageous in toxicology as environmental toxins (drugs) induce ferroptosis in healthy organs like the liver, kidneys, heart and brain, leading to toxic side effects. Biomarkers of ferroptosis can then help assess which organs are at risk of damage, allowing preemptive measures to protect these tissues.

Certain chemicals and pollutants, like heavy metals and industrial pollutants, directly or indirectly trigger ferroptosis. By measuring biomarkers of ferroptosis in animal models or cell cultures exposed to these substances, researchers can identify chemicals that pose significant health risks and implement safer guidelines for exposure levels. Biomarkers also serve as early indicators of ferroptosis-related damage in non-cancerous tissues. For example, if elevated levels of lipid peroxidation markers are detected in the blood following chemotherapy or toxins exposure, it could indicate liver or kidney toxicity, allowing clinicians to modify the treatment regimen to mitigate damage. When ferroptosis biomarkers suggest unwanted toxicity in healthy cells, ferroptosis inhibitors like ferrostatin-1, liproxstatin-1 or vitamin E could be utilized to mitigate tissue damage. Thus, biomarkers may be helpful to guide when and where these inhibitors are needed to protect healthy organs or cells. Several biomarkers of oxidative stress are biomarkers of ferroptosis such as glutathione depletion, GPX4 (glutathione peroxidase 4) inhibition, or formation of lipid peroxidation products. Our recent untargeted metabolomics studies showed that opthalamate levels were significantly elevated in human ovarian tumor cells undergoing ferroptosis [70]. Opthalamate levels have been shown to be elevated in during oxidative stress caused by the depletion of glutathione in liver [71]. Additionally, several genes, known markers of oxidative stress (e.g., CHAC1 and heme oxygenase), have been shown to be elevated during ferroptosis [72,73,74].

Specific biomarkers of ferroptosis are particularly valuable in toxicity research. Lipid peroxidation products (e.g., MDA, 4-HNE) are good indicators of lipid peroxidation and oxidative damage that can indicate ongoing ferroptosis. Iron dysregulation is another key feature of ferroptosis and measuring intracellular iron levels or proteins like ferritin and transferrin can indicate ferroptosis risk. Glutathione depletion and inhibition of GPX4, an enzyme that prevents lipid peroxidation, are central to ferroptosis. Measuring these markers can help detect cells in the process of ferroptosis. Additionally, ferroptosis depends on cysteine uptake via the cystine/glutamate antiporter (system Xc^−^) to synthesize glutathione. Reduced system Xc^−^ activity can indicate a heightened risk of ferroptosis.

While most ferroptosis research focuses on cancer or pathological conditions (e.g., liver injury), several genetic biomarkers of ferroptosis have been identified in normal liver physiology, and these are critical for understanding of the basal regulation of ferroptosis, susceptibility of cells to oxidative stress and homeostatic control of iron and lipid metabolism. In normal tissue, ferroptosis is tightly regulated by the balance of iron metabolism, lipid peroxidation, and antioxidant systems. These genetic biomarkers of ferroptosis (like *GPX4*, *SLC7A11*, *FTH1*, *ACSL4*, *Nrf2*) are expressed in normal cells and contribute to cellular homeostasis, rather than actively driving cell death [75,76,77,78,79,80]. Table 1 shows the commonly expressed genetic biomarkers of ferroptosis in normal cells (liver, kidney, etc.).

Development of easy-to-use biomarker probes can then assess the degree of cellular stress and risk of ferroptosis from exposure to environmental toxins, drugs, or pollutants. By understanding how these exposures lead to ferroptosis, researchers can better understand the potential for organ damage and disease. By tracking ferroptosis biomarkers, toxicologists can measure cumulative oxidative stress and iron overload in tissues exposed to harmful substances over time. These data can guide interventions, such as antioxidant treatments or iron chelation therapies, to reduce the likelihood of ferroptosis-related damage in at-risk populations. Biomarkers can help detect the early stages of ferroptosis-related toxicity before clinical symptoms appear, offering a window for intervention. For example, in cases of heavy metal exposure, ferroptosis biomarkers could signal the need for treatments that counter oxidative stress or iron accumulation, preventing progression to severe tissue damage. In both chemotherapy and toxicology, understanding the presence and levels of ferroptosis biomarkers can inform the design of drugs aimed at modulating ferroptosis. In toxicology, they can assist in creating protective drugs that inhibit ferroptosis in response to environmental toxins.

For diseases associated with ferroptosis (e.g., neurodegenerative diseases, liver disease), biomarkers can guide therapeutic interventions that either promote or inhibit ferroptosis as needed. For example, in neurodegeneration, where excessive ferroptosis can be damaging, biomarkers could be used to develop and test drugs that protect neurons from ferroptotic cell death.

### 1.9. Ferroptosis Inhibitors in Therapeutic and Preventive Measures

Research into ferroptosis has highlighted potential therapeutic strategies, including ferroptosis inhibitors like ferrostatin-1 and liproxstatin-1, which can prevent lipid peroxidation (Figure 2). These inhibitors might be used to mitigate the effects of pollutant-induced ferroptosis in high-risk populations. Supplementing with antioxidants (e.g., vitamin E, selenium) may help prevent lipid peroxidation, offering a potential preventive measure against ferroptosis for individuals exposed to environmental pollutants. Additionally, monitoring iron intake and iron levels could reduce susceptibility to ferroptosis in populations at risk.

It should be noted that several synthetic and natural compounds have been identified that act as inhibitors of ferroptosis. These include Iron chelators (e.g., deferoxamine) which reduce the availability of iron, a catalyst for lipid peroxidation [81]. Since GPX4 detoxifies lipid peroxides, and its activity is crucial to preventing ferroptosis GPX4 activators [82] and mimetics such as GPX4 Activator 2 (directly binds to GPX4 to activate its activity) or tannic acid (a natural polyphenol which activates and increases GPX4 levels [83]) could be utilized to inhibit ferroptosis.

However, despite the efficacy of synthetic inhibitors in experimental settings, their long-term safety and metabolic impacts are not known and is of concern. This has led to use of natural ferroptosis inhibitors, that are safer, and more suitable alternatives. Natural compounds such as Coenzyme Q10 (CoQ10), flavonoids, and polyphenols have gained interest due to their ability to inhibit ferroptosis, primarily through antioxidant and iron-chelating properties. CoQ10 is a lipid-soluble antioxidant present in mitochondrial membranes, and it plays a critical role in the electron transport chain and neutralization of lipid peroxides. It directly prevents ferroptosis by acting as a lipophilic antioxidant, inhibiting lipid peroxidation [84,85]. It enhances mitochondrial function and supporting endogenous antioxidant systems like GPX4 to reduce reactive ROS.

Flavonoids are polyphenolic compounds present in fruits, vegetables, and certain tea. They modulate ferroptosis through iron chelation, preventing Fenton reactions that generate harmful ROS. In addition, they are free radical scavengers and, inhibit lipid peroxidation. Polyphenols, including resveratrol, curcumin, and epigallocatechin gallate (EGCG), are potent antioxidants found in plant-based foods and have shown to suppress lipid ROS accumulation, offering neuroprotective and anti-aging benefits via ferroptosis inhibition [86]. Polyphenols have been reported to regulate redox-sensitive transcription factors like Nrf2, which induces antioxidant gene expression. A diet rich in antioxidant-rich, plant-based foods can act as a first line of defense, neutralizing ROS generated by environmental toxins.

Thus, ferroptosis inhibitors, especially natural compounds like CoQ10, flavonoids, and polyphenols, may be promising tools for both therapeutic and preventive medicine. Their role in decreasing oxidative damage, resulting from environmental pollutants, underscores the importance of a diet rich in antioxidants and phytochemicals. Promoting healthy eating habits not only supports cellular defense systems but also reduces the risk of ferroptosis-related damages, making nutrition an important strategy in combating modern environmental health challenges.

## 2. Conclusions and Future Developments

While significant progress is being made in the understanding of environmental pollutant-induced toxicity to humans, there is still a lot of research needed to further our understanding on the long-term effects of these pollutants. Furthermore, we do not know what effects of combinations of these pollutants have on human health, e.g., effects of PFAS with low doses of heavy metal or particulates combined with heavy metals.

There are a number of steps that must be taken in order to be more effective in discovering biomarkers and utilizing these biomarkers for effective interventions to help support long-term effects on humans. One way is to combine high-throughput screening and developing cost-effective assays of non-invasive biomarkers. In addition, implementing cost-effective validation strategies and collaborating across fields, we can conduct ferroptosis biomarker discovery in a budget-conscious manner. These approaches should help in streamlining processes, prioritize high-impact biomarkers, and reduce redundancy, paving the way for affordable, scalable research in both toxicology and chemotherapy.

There is a strong need for proper models to identify biomarkers of ferroptosis. Currently, the model for ferroptosis, human or animal cell lines are utilized to screen for ferroptosis biomarkers in response to toxicants which are more cost-effective than in vivo animal models. Well-characterized cell lines sensitive to ferroptosis can provide relevant insights at a fraction of cost. However, it has been found that such models (2D) differ from modifications in mice or humans in the process of drug screening and or discovery of biomarkers [87]. Using CRISPR/Cas9 or RNAi to knock down or knock out ferroptosis-related genes in cell models can help rapidly identify potential biomarkers without the need for more expensive or labor-intensive techniques.

With the recent advancement in 3D culture technology, normal (non-tumor), tumor organoid culture technology could be utilized which may be beneficial for discovery of biomarkers to prevent/mitigate diseases. Compared with traditional 2D culture and tumor tissue xenotransplantation, organoid models have a significantly higher success rate [87]. Other models have also been recently proposed to evaluate biomarkers of ferroptosis [88,89]. If animal models are necessary, zebrafish may be the most cost-effective alternative for studying toxicology and ferroptosis, given their low maintenance costs, rapid development, and high genetic homology with humans for relevant pathways [90].

## 3. Summary

Ferroptosis’s link to environmental pollutants has provided new understanding how toxicants can cause organ damage, neurotoxicity, and chronic disease. This cell death pathway highlights the importance of iron metabolism and lipid peroxidation in toxicology, as it offers potential therapeutic targets to reduce the impact of pollutants. Ferroptosis serves as a critical nexus between environmental exposures and cellular dysfunction. Recognizing the role of environmental pollutants in triggering ferroptosis not only enhances our understanding of disease pathogenesis but also guides the development of targeted interventions to protect public health. With increasing industrialization and repeated exposures to these toxicants and particulate matters, it is more critical to understand, evaluate and find correct solutions if we are to maintain a healthy lifestyle today and for future generations.

Nuclear factor erythroid 2–related factor 2 (NRF2), a transcription factor, plays a critical role in regulating cellular defenses against oxidative stress induced by environmental toxins such as per- and polyfluoroalkyl substances (PFAS) and heavy metals [35]. NRF2 is central to maintaining cellular homeostasis by modulating antioxidant responses, detoxification pathways, metabolism, and inflammation. It regulates various antioxidant genes, including the cystine/glutamate antiporter (system Xc^−^), specifically SLC7A11, which facilitates cystine uptake for glutathione (GSH) synthesis [79]. Since glutathione peroxidase 4 (GPX4) requires GSH to detoxify lipid peroxides and prevent ferroptosis [91], NRF2 activation is generally associated with ferroptosis inhibition [92]. However, the role of NRF2 in ferroptosis is context-dependent and complex. In some cases, NRF2 upregulation promotes ferroptosis. For instance, in lung cancer cells, elevated NRF2 levels have been shown to sensitize tumor cells to ferroptosis by increasing multidrug resistance-associated protein 1 (MRP1) expression. Moreover, NRF2-driven upregulation of ABCC1 (MRP1) can deplete intracellular GSH, further contributing to ferroptosis in tumor cells [93].

However, under conditions of prolonged or high-intensity exposure, NRF2′s role becomes more complex and potentially detrimental. Sustained NRF2 activation can induce expression of genes such as HO-1, leading to the production of biliverdin and the release of redox-active iron, which triggers lipid peroxidation in neuroblastoma and fibrosarcoma cells. Furthermore, excessive or chronic oxidative stress may surpass the capacity of NRF2-mediated defenses, particularly if glutathione is depleted or if GPX4 activity is impaired, leading to ferroptotic cell death despite continued NRF2 signaling. This dual nature of NRF2 also has significant implications in chemical risk assessment and environmental pollutants-response modeling. It clearly suggests the importance of exposure duration, dose, and cellular context in determining whether NRF2 serves a protective or ferroptosis-inducer. Additionally, genetic or epigenetic alterations in NRF2 or its regulatory network may determine individual variability in pollutants toxicity. Thus, modulating NRF2 or ferroptosis pathways may offer therapeutic strategies against specific pollutant, e.g., using ferroptosis inhibitors in the context of pollutant-induced environmental toxicity. Understanding the context-dependent role of NRF2 in ferroptosis adds an important layer to toxicological evaluations, where redox balance, iron metabolism, and regulated cell death pathways meet.

While the specific effects of PFAS on system Xc^−^ remain unclear, certain heavy metals are known to inhibit SLC7A11 and induce ferroptosis. PFAS are known inducers of oxidative stress and may potentially impair GSH synthesis by inhibiting SLC7A11, thereby affecting system Xc^−^ activity. A proposed pathway for environmental pollutant-induced ferroptosis is shown in Figure 2.

## Figures and Tables

**Figure 1 ijms-26-06658-f001:**
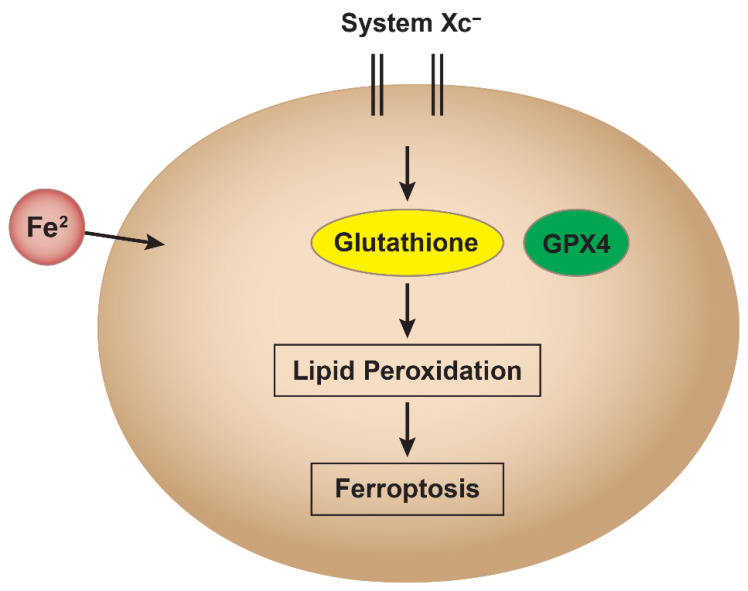
Schematic representation of the ferroptosis pathway. Loss of iron homeostasis and depletion of glutathione (GSH), caused by inhibition of system Xc^−^ and glutathione peroxidase 4 (GPX4) by chemicals or pollutants, lead to increased reactive oxygen species (ROS) formation. This oxidative stress results in lipid peroxidation, culminating in ferroptotic cell death.

**Figure 2 ijms-26-06658-f002:**
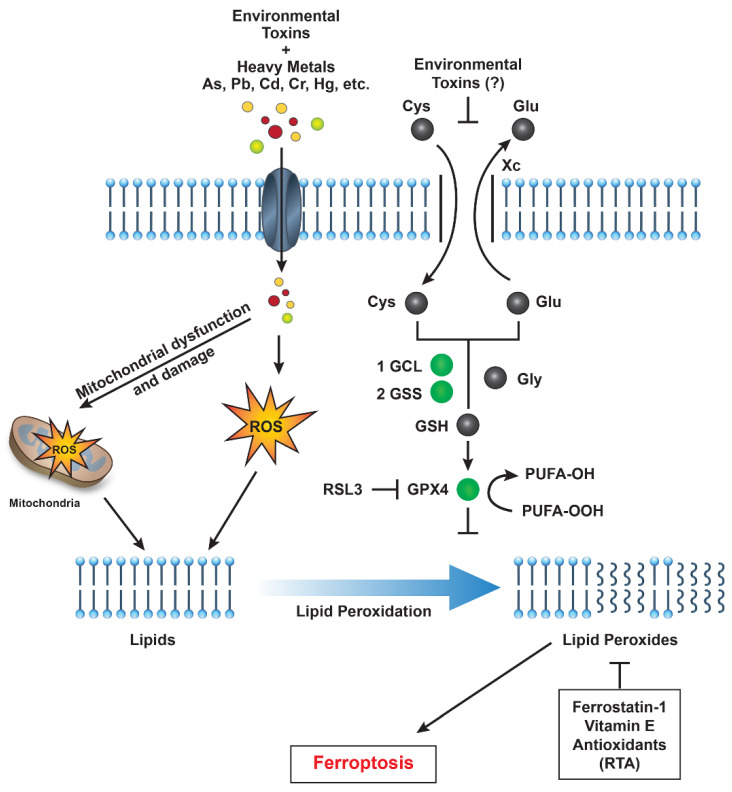
A schematic representation illustrating how environmental toxins—such as PFAS, heavy metals, and particulate matter—impact various cellular pathways that promote reactive oxygen species (ROS) generation, lipid peroxidation, and ultimately induce ferroptotic cell death in humans. Abbreviations: GCL (glutamate-cysteine ligase), GSS (glutathione synthetase), GPX4 (glutathione peroxidase 4), and RSL3 (Ras-selective lethal 3).

**Table 1 ijms-26-06658-t001:** List of commonly expressed genes, and their functions in normal tissues/cells.

Gene	Function	Expression in Tissue	Function in Ferroptosis
*GPX4*	Reduces lipid peroxides	All tissues	Inhibits ferroptosis
*SLC7A11*	Xc^−^ transporter	Neural, epithelial, immune	GSH synthesis, anti- ferroptosis
*ACSL4*	PUFA incorporation	Liver, Brain, testis	Sensitizes cells to ferroptosis
*FTH1*	Ferritin Heavy Chain1, iron storage	Liver, spleen, heart, brain	Sequesters iron, anti-ferroptosis
*FTL*	Ferritin Light Chain	Liver, spleen, heart, brain	Sequesters iron, anti-ferroptosis
*NCOL4*	Ferritin Cargo Receptor	Liver, gut, kidney, epithelial	Releases iron by ferritin degradation, pro-ferroptosis
*ALOX15*	Lipoxygenase, catalyzes lipid peroxidation	Lung, brain, Immune	Promotes lipid peroxidation, pro-ferroptosis
*TFRC*	Transferrin Receptor, iron uptake	Gut, kidney, proliferative tissues	Increases iron uptake, pro-ferroptosis
*NrF2*	Oxidative stress response	Lung, brain, liver	Activates antioxidant genes, anti-ferroptosis

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
