# Peer review of "Ferroptosis in Toxicology: Present and Future"

_ijms, 2025, doi:10.3390/ijms26146658_

Round 1

Reviewer 1 Report

Comments and Suggestions for Authors

This review represents a significant effort to integrate ferroptosis research with environmental toxicology, highlighting its potential to advance understanding of pollutant-induced cellular damage. By incorporating enhanced mechanistic details, rigorously validated figures, and clearer translational implications, the manuscript can solidify its position as a foundational resource in the field. However, several questions should be addressed before it is accepted.

  • Perfluoroalkyl and Polyfluoroalkyl Substances (PFAS):The statement "PFAS inhibit GPX4" requires a more detailed mechanistic explanation of the molecular interactions involved, including potential binding mechanisms or post-translational modifications.
  • Ferroptosis Combined with Apoptosis and Necrosis: Synergistic Effects of Ferroptosis with Apoptosis and Necrosis: The current discussion of "synergistic effects of multi-pollutant co-exposure" needs expansion with specific molecular pathways (e.g., crosstalk between ferroptosis-associated lipid peroxidation and apoptotic signaling) and supporting experimental evidence from relevant studies.
  • Ferroptosis Inhibitors in Therapeutic and Preventive Measures: NRF2's Dual Role: The discussion should be enhanced to address NRF2's context-dependent regulatory effects in ferroptosis, including its antioxidant protective functions versus its potential pro-ferroptotic effects under sustained oxidative stress conditions.
  • Figures and Data Support: The authors should provide a comprehensive mechanistic diagram illustrating: The pathways through which environmental toxins (PFAS, heavy metals, particulate matter) induce ferroptosis via System Xc⁻ inhibition, GSH depletion, and GPX4 suppression; Key molecular markers (e.g., ACSL4, TfR1) and potential intervention nodes (e.g., ferrostatin-1, vitamin E) with proper annotation
  • Format Consistency: Correct incomplete references (e.g., Ref. 1: Add journal volume number and page range); Clarify Ref. 60 by specifying the experimental models (cell types) and methodological approaches used in *Kirkwood-Donelson et al., 2024*

Author Response

This review represents a significant effort to integrate ferroptosis research with environmental toxicology, highlighting its potential to advance understanding of pollutant-induced cellular damage. By incorporating enhanced mechanistic details, rigorously validated figures, and clearer translational implications, the manuscript can solidify its position as a foundational resource in the field. However, several questions should be addressed before it is accepted.

  • Perfluoroalkyl and Polyfluoroalkyl Substances (PFAS): The statement "PFAS inhibit GPX4" requires a more detailed mechanistic explanation of the molecular interactions involved, including potential binding mechanisms or post-translational modifications.

This is included in the revised manuscript.

  • Ferroptosis Combined with Apoptosis and Necrosis: Synergistic Effects of Ferroptosis with Apoptosis and Necrosis: The current discussion of "synergistic effects of multi-pollutant co-exposure" needs expansion with specific molecular pathways (e.g., crosstalk between ferroptosis-associated lipid peroxidation and apoptotic signaling) and supporting experimental evidence from relevant studies.

This has been extended as suggested by the reviewer.

  • Ferroptosis Inhibitors in Therapeutic and Preventive Measures: NRF2's Dual Role: The discussion should be enhanced to address NRF2's context-dependent regulatory effects in ferroptosis, including its antioxidant protective functions versus its potential pro-ferroptotic effects under sustained oxidative stress conditions.

Has been revised to and included in the revised manuscript.

  • Figures and Data Support: The authors should provide a comprehensive mechanistic diagram illustrating: The pathways through which environmental toxins (PFAS, heavy metals, particulate matter) induce ferroptosis via System Xc⁻ inhibition, GSH depletion, and GPX4 suppression; Key molecular markers (e.g., ACSL4, TfR1) and potential intervention nodes (e.g., ferrostatin-1, vitamin E) with proper annotation

               I have added a new figure for this. Figure-1 and 2 contain most of these.

  1. Format Consistency: Correct incomplete references (e.g., Ref. 1: Add journal volume number and page range); Clarify Ref. 60 by specifying the experimental models (cell types) and methodological approaches used in *Kirkwood-Donelson et al., 2024*

References are in correct format for IJMS. Have clearified Kirkwood-Donelson reference and untargeted metabolomic and cell type (ovarian cancer cells) are added.

Reviewer 2 Report

Comments and Suggestions for Authors

Certain environmental and chemical toxins can induce ferroptosis either by disrupting iron homeostasis or by depleting glutathione, thereby enhancing reactive oxygen species (ROS) accumulation. Heavy metals like arsenic and cadmium, or compounds that inhibit GPX4, can trigger ferroptotic cell death, contributing to tissue injury and disease pathology. Understanding the interplay between toxins and ferroptosis is critical for developing therapeutic strategies against toxin-induced organ damage and ferroptosis-related diseases. In this review, author discussed this form of regulated cell death as an important mechanism to explain the toxicity of various environmental pollutants like heavy metals, pesticides, etc. Overall, this is a very well written review article first explaining the basic knowledge of ferroptosis and then discuss its role in toxicity potential of several environmental pollutants. There is one suggestion to further improve this review article- 1. Discuss more on ferroptosis inhibitors in therapeutic measures and may include the importance of natural products like CoQ10, flavonoids and polyphenols as ferroptosis inhibitors in preventive measures highlighting the benefits of eating healthy to mitigate the toxic effects of environmental pollutants.

Author Response

Certain environmental and chemical toxins can induce ferroptosis either by disrupting iron homeostasis or by depleting glutathione, thereby enhancing reactive oxygen species (ROS) accumulation. Heavy metals like arsenic and cadmium, or compounds that inhibit GPX4, can trigger ferroptotic cell death, contributing to tissue injury and disease pathology. Understanding the interplay between toxins and ferroptosis is critical for developing therapeutic strategies against toxin-induced organ damage and ferroptosis-related diseases.

In this review, author discussed this form of regulated cell death as an important mechanism to explain the toxicity of various environmental pollutants like heavy metals, pesticides, etc. Overall, this is a very well written review article first explaining the basic knowledge of ferroptosis and then discuss its role in toxicity potential of several environmental pollutants.

There is one suggestion to further improve this review article- 1. Discuss more on ferroptosis inhibitors in therapeutic measures and may include the importance of natural products like CoQ10, flavonoids and polyphenols as ferroptosis inhibitors in preventive measures highlighting the benefits of eating healthy to mitigate the toxic effects of environmental pollutants.

As suggested, we have now included the roles of CoQ10, flavonoids and polyphenols as modulators of ferroptosis and oxidative stress in the revised manuscript.

Reviewer 3 Report

Comments and Suggestions for Authors

This is a concise, clearly written review of ferroptosis. The author is not a major player in ferroptosis research, but even so he manages to produce a rather balanced and informative manuscript. In my view, the main shortcoming is the lack of a clear mechanism through which lipid peroxidation gives rise to ferroptosis, i.e. the curved arrow in Fig. 1, whose meaning should be detailed, at least in the form of hypotheses, with possible future experiments.

Author Response

I'm uncertain how to interpret the reviewer's comment—whether it's meant as a backhanded compliment suggesting that, while I may not be a major figure in the ferroptosis field, I can still put together a solid, balanced review; or as a subtle insult implying that my contributions are not significant. Regardless, I found the tone of the comment unprofessional. Reviewers should strive to maintain a respectful and objective tone throughout their evaluations.

For context, I am now 80 years old, retired for 25 years, and getting up in the morning to do many things that I like to do, including going to my lab, is a major accomplishment. I must have reviewed over 500 manuscripts throughout my career. In all that time, I have never submitted a review containing unprofessional remarks.

In the early 1980s, my group at the National Cancer Institute was investigating the mechanisms of Adriamycin-induced cell death in cancer cells, focusing on ROS-induced pathways. We observed significant lipid peroxidation as a result of Adriamycin treatment. At the time, we simply called this “lipid peroxidation”, as iron was considered to be fully bound—despite our awareness that iron was necessary for lipid peroxidation to occur. We utilized the MDA assay with TBA to quantify peroxidation, a method that remains in use today.

Regarding the current manuscript: in the original Figure 1 (now Figure 2), a curved arrow was used due to space constraints. This has now been replaced with a straight arrow. Additionally, I have included a simplified introductory figure (Figure-1) to illustrate the concept of ferroptosis more clearly.

Reviewer 4 Report

Comments and Suggestions for Authors

The manuscript FERROPTOSIS IN TOXICOLOGY: PRESENT AND FUTURE claims to be a review of the role of ferroptosis in toxicology. It essentially considers lipid peroxidation as one of the ferroptosis-related markers, but does not make a connection between the main pathways regulating ferroptosis and the toxicological response at the cellular level.
The second section of the manuscript is entitled Perfluoroalkyl and Polyfluoroalkyl Substances (PFAS), but it also includes information on other pollutants such as heavy metals, pesticides, air pollutants and their role in oxidative stress and glutathione metabolism. Thus, the title does not correspond to the content of this section. The following third section, which is related to ferroptosis combined with apoptosis and necrosis, is extremely short and uninformative (9 lines in total).
The section Biomarkers of ferroptosis discusses biomarkers of oxidative stress and glutathione metabolism, and not the genetic markers specific to this programmed cell death. A detailed and good review on this issue can be seen at https://www.nature.com/articles/s41419-020-2298-2
At the end, an attempt is made to review ferroptosis inhibitors in therapies and prevention, but again brief and uninformative in essence.

Author Response

Genetic markers differ between normal and cancer cells. While the majority of biomarker research has focused on cancer cells, relatively little work has been conducted on normal cells or tissues. Moreover, different cell types exhibit distinct genetic biomarkers. In response to this, I have now included a new table in the revised manuscript listing common ferroptosis biomarkers expressed in normal cells.

 The reference suggested by this reviewer is already cited in the review as Reference 9.

The section on ferroptosis inhibitors in therapy and prevention has been expanded to include a broader range of compound classes, as recommended by Reviewer 2.

Round 2

Reviewer 4 Report

Comments and Suggestions for Authors

The revised version of the manuscript has been significantly improved. Most of the shortcomings have been eliminated and it has significantly greater informative value compared to the first version. In this form, the manuscript can be accepted for publication. A few more corrections are needed: All subsections in point 1 are numbered as 1.1 (a total of two points 1 and a total of five subsections 1.1). It is not clear what is the logic of heavy metals, industrial pollutants, pesticides being defined as a subsection of Teflon pollutants, such as Perfluoroalkyl and Polyfluoroalkyl Substances (PFAS)? I would also title the subsections of the point 1. Ferroptosis Combined with Apoptosis and Necrosis as 2.1 Ferroptosis and apoptosis crosslinks instead of interaction (the same applies to the connection between ferroptosis and necrosis). These points should be expanded upon with a discussion of general signaling mechanisms and regulators, as well as what determines which direction the cell will take - necrosis, apoptosis, or ferroptosis.

Author Response

Ferroptosis and apoptosis crosslink and ferroptosis and necrosis crosslinks have been expanded to include reviewer’s comments regarding signaling mechanisms and regulators. Additional references have been included in this revised manuscript.